# Neutrophil Extracellular Traps in Severe SARS-CoV-2 Infection: A Possible Impact of LPS and (1→3)-β-D-glucan in Blood from Gut Translocation

**DOI:** 10.3390/cells11071103

**Published:** 2022-03-24

**Authors:** Supichcha Saithong, Navaporn Worasilchai, Wilasinee Saisorn, Kanyarat Udompornpitak, Thansita Bhunyakarnjanarat, Ariya Chindamporn, Punyot Tovichayathamrong, Pattama Torvorapanit, Direkrit Chiewchengchol, Wiwat Chancharoenthana, Asada Leelahavanichkul

**Affiliations:** 1Department of Microbiology, Faculty of Medicine, Chulalongkorn University, Bangkok 10330, Thailand; supichcha_mumt@hotmail.com (S.S.); wsaisorn@gmail.com (W.S.); jubjiibb@hotmail.com (K.U.); thansitadew@gmail.com (T.B.); ariya.c@chula.ac.th (A.C.); punyot.tovi@gmail.com (P.T.); bee982@hotmail.com (D.C.); 2Center of Excellence on Translational Research in Inflammation and Immunology (CETRII), Department of Microbiology, Chulalongkorn University, Bangkok 10330, Thailand; 3Department of Transfusion Medicine and Clinical Microbiology, Faculty of Allied Health Sciences, Chulalongkorn University, Bangkok 10330, Thailand; navaporn.w@chula.ac.th; 4Immunomodulation of Natural Products Research Group, Chulalongkorn University, Bangkok 10330, Thailand; 5Thai Red Cross Emerging Infectious Diseases Clinical Center, King Chulalongkorn Memorial Hospital, Bangkok 10330, Thailand; ptorvorapanit@gmail.com; 6Tropical Nephrology Research Unit, Department of Clinical Tropical Medicine, Faculty of Tropical Medicine, Mahidol University, Bangkok 10400, Thailand; 7Tropical Immunology and Translational Research Unit, Department of Clinical Tropical Medicine, Faculty of Tropical Medicine, Mahidol University, Bangkok 10400, Thailand; 8Nephrology Unit, Department of Medicine, Faculty of Medicine, Chulalongkorn University, Bangkok 10330, Thailand

**Keywords:** COVID-19, NETs, endotoxemia, glucanemia, gut barrier defect, blood bacteriome

## Abstract

Due to limited data on the link between gut barrier defects (leaky gut) and neutrophil extracellular traps (NETs) in coronavirus disease 2019 (COVID-19), blood samples of COVID-19 cases—mild (upper respiratory tract symptoms without pneumonia; *n* = 27), moderate (pneumonia without hypoxia; *n* = 28), and severe (pneumonia with hypoxia; *n* = 20)—versus healthy control (*n* = 15) were evaluated, together with in vitro experiments. Accordingly, neutrophil counts, serum cytokines (IL-6 and IL-8), lipopolysaccharide (LPS), bacteria-free DNA, and NETs parameters (fluorescent-stained nuclear morphology, dsDNA, neutrophil elastase, histone–DNA complex, and myeloperoxidase–DNA complex) were found to differentiate COVID-19 severity, whereas serum (1→3)-β-D-glucan (BG) was different between the control and COVID-19 cases. Despite non-detectable bacteria-free DNA in the blood of healthy volunteers, using blood bacteriome analysis, proteobacterial DNA was similarly predominant in both control and COVID-19 cases (all severities). In parallel, only COVID-19 samples from moderate and severe cases, but not mild cases, were activated in vitro NETs, as determined by supernatant dsDNA, *Peptidyl Arginine Deiminase 4*, and nuclear morphology. With neutrophil experiments, LPS plus BG (LPS + BG) more prominently induced NETs, cytokines, *NFκB*, and reactive oxygen species, when compared with the activation by each molecule alone. In conclusion, pathogen molecules (LPS and BG) from gut translocation along with neutrophilia and cytokinemia in COVID-19-activated, NETs-induced hyperinflammation.

## 1. Introduction

A novel β-coronavirus, from the severe acute respiratory syndrome (SARS) coronavirus (CoV) family, referred to as “SARS-CoV-2”, is the cause of the coronavirus disease 2019 (COVID-19) infection [1]. Different from most respiratory viral infections, where the viral entry is limited only to specific cells in the oral–nasopharyngeal–respiratory epithelium, SARS-CoV-2 also infects several other organs (lung, stomach, small intestine, colon, skin, lymph node, thymus, bone marrow, spleen, liver, kidney, and brain) through angiotensin-converting enzyme 2 (ACE2) receptor [2]. Perhaps, the capability of COVID-19 in widespread infection into several cell types, including the immune cells, might—at least in part—result in the profound cytokine in the systemic circulation (cytokine storm) from cytokine production of different cell types in the host [3]. Additionally, COVID-19 not only induces cytokine storm but also causes neutrophilia [4], leading to the use of immunosuppressive treatment [5] that is not common in other viral infections [6]. Indeed, COVID-19-induced neutrophilia might be associated with profound systemic inflammatory cytokines during COVID-19 infection through several proinflammatory activities, including neutrophil extracellular traps (NETs) [7]. As such, NETosis (neutrophils death after NETs) is inducible by several mechanisms, including damage-associated molecular patterns (DAMPs) or pathogen-associated molecular patterns (PAMPs), through the peptidyl-arginine deiminase 4 (*PAD4*) enzyme, that can be activated by the NADPH oxidase 2 (NOX2)-dependent or NOX2-independent pathway [8,9]. During severe infection, both DAMPs and PAMPs activate NETs formation that partly enhances cytokine storm and systemic inflammation, leading to the prominent multi-organ dysfunctions, referred to as “sepsis” [10]. Accordingly, sepsis is a life-threatening organ failure caused by systemic infection, regardless of the type of organisms (bacteria, viruses, and fungi) [11]. Among viral infections, sepsis cytokine storm caused by COVID-19, dengue, influenza, and respiratory syncytial virus (RSV) is well known, and possibly associated with the extensiveness of host cells that can be infected and the quantity of the cytokines that are produced from these infected cells [12].

In a severe viral infection, diarrhea could be caused by a direct enterocyte virus infection (such as RSV) [13] and/or sepsis-associated factors (cytokinemia, endotoxemia, and gut dysbiosis), as demonstrated in severe dengue infection [14,15]. As such, hypercytokinemia in sepsis induces the breakdown of intestinal tight junctions (TJs), allowing the translocation of PAMPs from the gut into blood circulation (gut translocation) [16,17,18,19,20,21]. Notably, the gut barrier between the host and GI contents is formed only by a single layer of epithelial cells with an approximate surface area of 32 m^2^, held together by TJs, forming an intrinsic mucosal defense system [22]. Loss of the TJ barrier allows translocation of lipopolysaccharide (LPS) and (1→3)-β-D-glucan (BG)—the most and second most present PAMPs in feces, respectively [9,22,23] from the gut into the blood circulation (gut translocation), further enhancing systemic inflammation [24]. Both LPS and BG are the most important cell wall structures of Gram-negative bacteria and fungi, respectively, that synergistically facilitate inflammatory responses and NETosis, partly through Toll-like receptor-4 (TLR-4) and Dectin-1, respectively (9). Although endotoxemia, positive blood bacterial DNA (DNAemia) [25,26], and blood bacteriome [27] in COVID-19 cases without bacterial infection have implied a gut barrier defect (leaky gut), data on this topic is still lacking. Furthermore, the simultaneous presence of LPS and BG in COVID-19 infection with neutrophil predominance might easily enhance NETosis and causes more severe sepsis. We hypothesized that severe COVID-19 (direct enterocyte infection and systemic cytokines) causes leaky gut and gut translocation of PAMPs that facilitate NETosis and sepsis severity, for the following reasons: (i) because direct enterocyte infection by SAR-CoV-2 through ACE-2 and the presence of diarrhea in severe COVID-19 that might be associated with TJs defect [28]; and (ii) because of the associations between COVID-19-activated factors (cytokine storm, neutrophilia, endotoxemia, and NETosis) [29] and gut barrier defect, which are mentioned in [22,30,31]. Hence, several parameters were evaluated from blood samples of patients with different COVID-19 severity, and the isolated neutrophils were stimulated in vitro by patient-derived serum or LPS, with or without BG, to determine a linkage between pathogen molecules and NETosis in COVID-19 infection.

## 2. Materials and Methods

### 2.1. Participants Enrollement and Study Designs

Confirmed cases of COVID-19—determined to be positive for SARS-CoV-2 RNA using real-time reverse transcription polymerase chain reaction (RT-PCR) test from combined nasopharyngeal and throat swab samples—were recruited in the Department of Microbiology, Faculty of Medicine at Chulalongkorn University, according to the approved Institutional Review Board of the Faculty of Medicine (IRB No. 426/63, COA No. 738/2020). Samples from COVID-19-positive patients between 18 and 70 years old during March–May 2021 were collected on admission (moderate and severe cases) or at the first hospital examination (mild cases). Due to the higher incidence of patients with mild disease severity, 1 of every 2 samples from this group were randomly selected, while all samples from moderate and severe cases were used. Because of the possible confounding factors on endotoxemia and glucanemia, patients with the following exclusion criteria were excluded: other infections (bacteria and fungi with either local or systemic sources); kidney or liver injury (serum creatinine higher than 1.8 mg/dL or alanine transaminase higher than 60 U/L); significant hemodynamic instability (mean arterial pressure lower than 60 mmHg); and vasopressor (or antibiotics) administration. SARS-CoV-2 RNA detection was performed using Cobas SARS-CoV-2 assay on an automated Cobas 6800 system (Roche Diagnostics, Basel, Switzerland), during which the nucleic acid was automatically extracted from 400 μL of specimens, following a previous publication [32]. The test is based on the detection of E and ORF1ab genes of SARS-CoV-2 and on cycle threshold (CT) values which, with amplification higher than 40 cycles, were determined as negative results. The highest CT value for COVID-19 diagnosis in this study was 36 cycles. Classification of the disease severity according to the World Health Organization (WHO) guidelines [33] was determined at the hospital visit or admission, as follows: (i) mild cases—COVID-19 without evidence of viral pneumonia or hypoxia; (ii) moderate cases—no signs of severe pneumonia, including SpO_2_ ≥ 90% on room air; (iii) severe cases—signs of pneumonia, including SpO_2_ < 90% on room air; and (iv) critical cases—acute respiratory distress syndrome (ARDS) with SpO_2_ ≤ 97%. Notably, critical cases were excluded from the study because the patient characteristics in most of the critical cases were interfered with by the features of sepsis. The epidemiologic data were obtained from hospital medical records. The healthy control group (15 plasma samples) was collected from volunteers, with signed informed consent. Blood samples of these patients were collected during May 2020.

### 2.2. Blood Sample Analyses

Complete blood count (CBC), serum creatinine, alanine transaminase (ALT), lactate dehydrogenase (LDH), and C-reactive protein (CRP) were measured in the central laboratory of the King Chulalongkorn Memorial Hospital using Sysmex XN9203 Analyzer (Kobe, Hyogo, Japan) for CBC and Cobas c502 (Roche Diagnostics, Basel, Switzerland) for other parameters. Procalcitonin and serum cytokines (TNF-α, IL-1β, IL-6, and IL-8) were measured by enzyme-linked immunosorbent assay (ELISA) (Invitrogen, Waltham, MA, USA). Serum LPS (endotoxin) and (1→3)-β-D-glucan (BG) were evaluated by HEK-Blue LPS Detection Kit 2 (InvivoGen™, San Diego, CA, USA) and Fungitell^®^ assay (Associates of Cape Cod, Falmouth, MA, USA), respectively. Due to the test’s lower-limit values, values less than 0.01 EU/mL and 7.8 pg/mL in LPS and BG assay, respectively, were recorded as 0. Briefly, 10^5^ cells/well of HEK-Blue cells, human embryonic kidney (HEK) cells, transfected with SEAP (secreted embryonic alkaline phosphatase) gene, were added into 96-well plates and incubated overnight at 37 °C in 5% CO_2_. After washing, 20 μL of serum and standard LPS and 180 μL of QUANTI-Blue were added to the wells and incubated for 1 h at 37 °C in 5% CO_2_. Then, the absorbance of each well was read at 620 nm by a fluorescent microplate reader (Bio-Tek, Winooski, VT, USA). For blood-bacteria-free DNA, DNA in serum was extracted with 5 M potassium acetate/acetic acid buffer and quantified by a Nanodrop 100 Spectrophotometer (Thermo Scientific, Paisley, UK). After that, the samples were examined using 16 s primer 5′-GATGAACGCTGGCGGCGTGC-3′ (F), 5′-CAATCATTTGTCCCACCTTC-3′ (R) by quantitative real-time polymerase chain reaction (PCR) with QuantStudio 6 Flex Real-time PCR System to determine bacterial DNA from the cell-free DNA preparations.

To determine neutrophil extra-cellular traps (NETs) from the blood samples, several parameters, including dsDNA, neutrophil elastase, histone–DNA complexes, and myeloperoxidase (MPO)–DNA complexes, were evaluated. For serum dsDNA, Quant-iT™ PicoGreen reagent (Thermo Fisher Scientific, Paisley, UK) was used according to the manufacturer’s directions. Briefly, an aqueous working solution was added to the samples for 5 min at room temperature and measured dsDNA at 480/520 nm on a fluorescent microplate reader (Bio-Tek, Santa Clara, CA, USA) [8,9,24]. Proinflammatory cytokines (TNF-α, IL-1β, IL-6, and IL-8) and neutrophil elastase were measured by enzyme-linked immunosorbent assay (ELISA) from Invitrogen (Waltham, MA, USA) and Abcam270204 (Cambridge, UK), respectively. In parallel, serum histone–DNA complexes were measured by photometric enzyme immunoassay using the Cell Death Detection ELISA (Roche, Basel, Switzerland). Meanwhile, MPO–DNA complexes were determined by the coated 96-flat-well plates after overnight incubation with 1 µg/mL anti-human MPO antibody (Bio-Rad 0400-0002, Oxford, UK) at 4 °C in the coating buffer (Abcam270204), as previously described [34,35,36]. Then, the samples were added and incubated at room temperature for 90 min before incubation by HRP-conjugated anti-DNA antibody (Abcam270204). After that, the color was developed with 3,3’,5,5’-Tetramethylbenzidine (TMB) substrate (Invitrogen) followed by 2 N H_2_SO_4_ for stop reaction and a wavelength of 450 nm by fluorescent microplate reader (Bio-Tek).

To identify sources of bacteria-free DNA in the blood, bacteriome analysis was performed using 200 µL of blood per sample, as previously described [31]. Briefly, metagenomic DNA was prepared from the samples using a QIAamp assay (Qiagen, Valencia, CA, USA) with DNA quality assessment using nanodrop spectrophotometry. Universal prokaryotic primers 515F (5′-GTGCCAGCMGCCGCGGTAA-3′) and 806R (5′-GGACTACHVGGGTWTCTAAT-3′) with appended 50 Illumina adapter and 30 Golay barcode sequences were used for 16S rRNA gene V4 library construction. Bioinformatic analyses were performed following Mothur’s standard operating procedures [31,37].

### 2.3. NETs Activation by Patient Serum, LPS, and BG

Neutrophils were isolated by density centrifugation with PolymorphprepTM (Axis-Shield, Oslo, Norway) and were resuspended in RPMI 1640 supplemented with 10% fetal bovine serum. Then, purified healthy neutrophils (5 × 10^5^ cells/well) were placed onto Poly-L- Lysine (Sigma-Aldrich, Singapore) coated on 6 mm glass coverslips and allowed to attach to the plate before challenging with 10% volume-by-volume (*v/v*) diluted serum from COVID-19 patients or healthy serum at 37 °C, 5% CO_2_ incubator. Additionally, Phorbol-12-Myristate-13-Acetate (PMA), a NETs inducer, at 30 ng/mL (Sigma-Aldrich) was used as a positive control. After 2 h of incubation, the coverslips were fixed with 4% formaldehyde, blocked with Tris-buffered saline (TBS) in 2% bovine serum albumin (BSA) (Sigma-Aldrich), and permeabilized by TBS with 0.05% Tween 20 (Sigma-Aldrich). Then, NETs formation was detected by nuclear morphology staining using a blue-fluorescent DNA-staining color 4′,6-diamidino2-phenylindole (DAPI) [8,9,24]. For detection of free dsDNA in the supernatant, the supernatant from 2-hour-stimulated neutrophils (5 × 10^5^ cells/well) was mixed with QuantiTTM PicoGreen reagent (Thermo Fisher Scientific). Expression of several genes was evaluated in relative to *β-actin* by RT-PCR using the following primers: PAD4 forward 5′-ACAGGTGAAAGCAGCCAGC-3′, reverse 5′-AGTGATGTAGATCAGGGCTTGG-3′; NFκB forward 5′-CTTCCTCAGCCATGGTACCTCT-3′, reverse 5′-CAAGTCTTCATCAGCATCAAACTG-3′, and β-actin forward 5′-GGACTTCGAGCAAGAGATGG-3′, reverse 5′-AGCACTGTGTTGGCGTACAG-3′. The PCR samples were prepared using an RNA-easy mini kit (Qiagen, Hilden, Germany), nanodrop 100 spectrophotometer, high-capacity cDNA reverse transcription, and SYBR Green PCR Master Mix for quantitative RT-PCR, with QuantStudio6 Flex Real-time PCR System (Thermo Scientific). The results were demonstrated in relative quantitation of the comparative threshold method (2^−ΔΔCt^) as normalized by β-actin. Because of the LPS and BG in the COVID-19 serum [38,39,40] and pathogen-molecule-activated NETs (9), the in vitro test was conducted on normal neutrophils. As such, neutrophils (5 × 10^4^ cells/well) were placed onto Poly-L-Lysine-coated (Sigma-Aldrich) glass coverslips, incubated at 37 °C, 5% CO_2_ with BG using whole glucan particle (WGP) (the purified BG from Saccharomyces cerevisiae) (Biothera, Eagan, MN, USA) at 10 μg/mL with or without LPS (Escherichia coli 026: B6) (Sigma-Aldrich), at 10 μg/mL for 2 h, before determination of supernatant dsDNA, *PAD-4* expression, and DAPI nuclear staining. In LPS-BG experiments, supernatant cytokines were also measured by ELISA (Invitrogen), and reactive oxygen species (ROS) production was determined by dihydroethidium (DHE) assay (Abcam, Cambridge, MA, USA), according to the manufacturer’s instructions. In brief, after cell culture preparation in 96-well plates, cell media was removed gently and 20 μM DHE was added to each well (50 μL) and incubated at 37 °C with 5% CO_2_ for 1 h. The plate was transfered to the fluorescent plate reader (Bio-Tek) and the fluorescence was measured using an excitation wavelength of 518 nm and an emission wavelength of 605 nm.

### 2.4. Statistical Analysis

All data were analyzed by Statistical Package for Social Sciences software (SPSS 22.0, SPSS Inc., Chicago, IL, USA) and Graph Pad Prism version 7.0 software (La Jolla, CA, USA). Results were presented as mean ± standard error (SE), except for the demographic data that used mean ± standard deviation (SD). The differences between multiple groups were examined for statistical significance by one-way analysis of variance (ANOVA) with Tukey’s analysis. A *p*-value < 0.05 was considered statistically significant.

## 3. Results

### 3.1. Demographic Data of the Population

In the sampled population, there were 105 confirmed cases of SARS-CoV-2 infection during March–May 2021, including 57 cases in the mild category (upper respiratory symptoms), 28 cases in the moderate category (pneumonia without hypoxia), and 20 cases in the severe category (pneumonia with hypoxia). In 57 mild cases, 1 out of 2 samples were randomly selected, with 27 total samples in the mild-severity group. The baseline clinical characteristics (mean ± standard deviation) of the patients and the healthy controls are demonstrated in Table 1. Patient age in the severe cases was higher than others, with males predominating in most groups, except for moderate-severity cases and control. All patients with severe infection had at least one underlying disease, while 63% of mild COVID-19 cases did not have any underlying diseases.

### 3.2. Prominent Neutrophilia and Neutrophil Lymphocyte Ratio in Severe COVID-19 Infection

With conventional blood parameters (Figure 1A–I), severe COVID-19 infection demonstrated the highest blood neutrophils and neutrophil/lymphocyte ratio (Figure 1B,D), but not the highest total white blood cell count (WBC), lymphocytes, platelet, organ injury (kidney and liver), lactate dehydrogenase (LDH), or C-reactive protein (CRP) (Figure 1A–I). The total WBC of the moderate-severity group was similar to the severe group (Figure 1A), while LDH and CRP in patients (all cases regardless of severity) were higher than the healthy control (Figure 1I).

In parallel, for the sepsis parameters (Figure 2A–H), serum TNF-α, IL-1β, IL-6, IL-8, endotoxemia, and blood-bacteria-free DNA in severe cases were higher than other groups and most of the levels of these parameters were different between COVID-19 severities. However, serum TNF-α, IL-1β, IL-6, and blood-bacteria-free DNA were not different between healthy control and mild COVID-19 cases (Figure 2B–D,H). Meanwhile, procalcitonin and serum (1→3)-β-D-glucan (BG) in the infected individuals were higher than the control but did not differ among infected groups (Figure 2A,G). Only serum TNF-α, IL-8, and endotoxemia could be distinguished between the infected cases versus healthy volunteers and the different severities of infection (Figure 2B,E,F).

### 3.3. Positive Bacteriome Analysis in Blood: Indirect Evidence of Gut Translocation

Because bacteria-free DNA could be demonstrated in the blood of moderate and severe COVID-19-infected cases (Figure 2F) without bacteremia or fungemia (data not shown), these bacterial DNA might originate from the intestines (an endogenous source of bacteria in the host) due to sepsis-induced gut barrier defect [30]. Notably, bacteria-free DNA in blood might originate from the translocation of the only bacteria-free DNA from the intestine or breakdown of the viable bacteria that transferred from the intestine, both of which are the results of gut translocation. Subsequently, blood bacteriome analysis was performed in the control versus the moderate and severe COVID-19 cases, but not in the mild cases, due to the undetectable bacteria-free DNA in the samples of the mild COVID-19 cases (Figure 2F). Despite non-detectable bacteria-free DNA in the healthy controls (Figure 2F), blood bacteriome was detectable in both the healthy control and COVID-19 cases (moderate and severe groups) with a similar category of bacteria, in which proteobacteria (Gram-negative bacteria with intestinal invasion capacity [38]) were the most predominant groups (Figure 3A–D). These data implied gut translocation in healthy individuals that were detectable only by bacteriome analysis, but not the conventional method supported the possible physiologic gut translocation [41,42]. However, the abundance of *E. coli* DNA in severe cases was higher than the moderate-severity group (Figure 3D), implying a possibly more prominent gut translocation of bacteria-free DNA in severe COVID-19 cases.

### 3.4. Serum from Severe COVID-19 Infection Prominently Induced NETosis in Patients and in Isolated Neutrophils from Healthy Volunteers

Because of the simultaneous elevation of pathogen molecules (LPS and BG) and neutrophilia (Figure 1B and Figure 2F,G), NETs formation was further evaluated with several parameters (Figure 4A–F). Indeed, fiber-like characteristics of the nuclei of neutrophils, blood dsDNA, and blood neutrophil elastase were most prominent in severe COVID-19 cases (Figure 4A–D). Meanwhile, the histone–DNA complex was similar between moderate and severe infection (higher than mild COVID-19 cases), and was significantly less in the control group (Figure 4E). Despite similar levels of the myeloperoxidase (MPO)–DNA complex between mild versus control and moderate versus severe COVID-19 cases, histone–DNA was higher in the moderate and severe COVID-19 cases than the mild cases and the control (Figure 4F). These data supported the idea that NETosis is enhanced in COVID-19 infection [29], possibly due to a simultaneous elevation of endotoxemia and glucanemia during COVID-19-induced neutrophilia (Figure 1B and Figure 2D,E). To test this hypothesis, neutrophils were isolated from the healthy individuals and activated by serum from the COVID-19 cases in comparison with the serum from healthy volunteers, using PMA as a positive control NETs inducer, as evaluated by several NETs parameters: supernatant dsDNA, *PAD-4* expression, and fluorescent-stained nuclear morphology (Figure 5A–D). Accordingly, serum from mild COVID-19 cases could not induce NETs, serum of moderate cases activated NETs to similar level as the PMA of the positive control, and severe case serum caused higher NETs than PMA (Figure 5A–D). Perhaps the combination of some blood components (such as cytokines, LPS, and BG) in serum from moderate and severe COVID-19 cases potently induced NETosis. Because our hypothesis focuses on an impact of gut barrier defect in COVID-1, and because pathogen molecules (both LPS and BG) are capable of NETs activation [9], activation by LPS and/or BG were tested in vitro (Figure 6A–D).

Indeed, both LPS and BG activated NETtosis, and combined LPS and BG (LPS + BG) enhanced NETs formation more prominently than the activation by either molecule alone; additionally, LPS + BG induced NETs less prominently than the PMA-positive control (Figure 6A–D). These data implied that gut translocation of LPS + BG was partially responsible for NETosis during severe COVID-19 infection. Because cytokines and reactive oxygen species (ROS) might be responsible for NETs formation, these parameters in LPS with or without BG activated neutrophils were also tested. Interestingly, PMA activated NETs without an impact on cytokine production and *NFκB* (a transcriptional factor) upregulation (Figure 6A–D), but elevated ROS production (Figure 6E–I), supporting PMA-induced NETs through ROS mediation [31]. Meanwhile, BG alone facilitated NETs (Figure 6A–D), and upregulated *NFκB*, but did not alter cytokines (Figure 6E–I), suggesting the non-cytokine-associated NETs formed after BG activation. On the other hand, LPS or LPS + BG potently activated inflammatory cytokines, especially TNF-α, along with *NFκB* upregulation and ROS production (Figure 6E–I), implying an influence of cytokine-associated NETosis, which might be a major mechanism of COVID-19-induced NETosis [43,44].

## 4. Discussion

The cross-sectional analysis showed that severe COVID-19 infection facilitated gut barrier defects, as indicated by the presence of LPS and BG in the blood, without systemic infections to enhance NETs formation and sepsis severity.

### 4.1. Neutrophilia in COVID-19 Infection

During SARS-CoV-2 infection, neutrophil accumulation is originally observed in the nasopharyngeal epithelium [45], lung [46], and subsequently in the blood (neutrophilia) [47], implying a prominent impact of neutrophils in the pathophysiology of COVID-19. In general, neutrophils are promptly recruited into the site of infection, and neutrophilia is common at the very early phase of viral infections [48,49]. Indeed, neutrophils are the first-line immune cells in the acute phase of most infections (bacteria, viruses, and fungi) [50,51] and other non-infectious inflammatory conditions [52]. In most viral infections, neutrophil numbers peak during the early phase of infections, follow by neutrophil reduction and lymphocyte elevation (lymphocytosis). However, the neutrophil recruitment that continues in a longer period during more severe infection (persistent neutrophilia), in which the neutrophil abundance is usually lower than the peak valve, is demonstrated in severe pneumonia from several viruses (such as SARS, influenza, and RSV) [49]. Although these neutrophils might effectively control viruses, they can also cause more severe organ injury, as indicated by pulmonary fibrosis after COVID-19 infection [53]. Among viral sepsis conditions, neutrophilia presents in severe respiratory infection (RSV, influenza, SARS, and COVID-19) [54,55,56], but not in severe dengue infection, which induces neutropenia with cytokinemia [57], which implies a diverse role of neutrophils in different viral sepsis conditions. While several types of host cells can be infected by COVID-19 (ACE-2 positive cells) and dengue viruses (leukocytes, liver, spleen, and endothelium), which both partly induce cytokine storm and NETosis [58,59,60,61], neutrophil-induced complications in COVID-19 might be more profound than dengue due to the higher blood neutrophil count. Indeed, we demonstrated that blood neutrophil count, serum IL-6, and serum IL-8 were the possible clinically correlated parameters of COVID-19 infection, which might be associated with sepsis severity; this is supported by previous observations [62,63]. Because of prominent neutrophilia in COVID-19 infection, NETosis might be more severe than other viral infections and might be associated with COVID-19 hyperinflammation [64].

### 4.2. Gut Barrier Defects in COVID-19

Without bacteremia and fungemia, the presence of LPS and BG in blood during COVID-19 infection implies gut translocation of these pathogen molecules that might be associated with hypercytokinemia and severe inflammation [38,65,66]. Although gut barrier defect (leaky gut) is also demonstrated in severe dengue infection (cytokine storm) [15,67], gut barrier defect in COVID-19 infection—especially in the cases with mild severity—without the detectable serum IL-6 might be due to the direct viral infection in enterocytes that possibly induce diarrhea [28]. Despite non-overt diarrhea in our patients, subtle diarrhea might be missing from our medical data records. Nevertheless, leaky gut in COVID-19 infection is severe enough for the translocation of LPS, BG, and, perhaps, fragmented bacterial DNA, but not the viable organisms. Indeed, the presence of bacteria-free DNA in serum without positive conventional bacterial culture could be due to the death of the viable bacteria after gut translocation (DNA from lytic dead bacteria) or the transfer of only bacteria-free DNA (but not the whole viable bacteria) through the gut barrier [68]. Notably, intact bacteria-free DNA (6.5 × 10^4^–9.8 × 10^6^ kDa) naturally breaks down into fragment-free DNA (less than 65 kDa) [69,70,71,72], similar to the size of LPS and BG (10–100 kDa) [73,74,75]; however, this could not pass through the healthy gut barrier, which only allows the passive transport of molecules smaller than 0.6 kDa [76,77]. Interestingly, despite the non-detectable blood-bacteria-free DNA, the existence of bacteriome in the blood of healthy volunteers suggests gut translocation of very small DNA fragments or a minor gut translocation of viable bacterial that could be effectively controlled by immune responses resulting in negative bacteremia. As such, DNA from proteobacteria—the groups of intestinal invasive bacteria [78]—in the blood bacteriome analysis of both healthy control and COVID-19 cases, suggests an impact of bacterial invasiveness on gut translocation because the DNA of the most prominent bacteria in feces (Firmicutes and Bacteroides) [79] were not dominant in blood bacteriome. Similarly, proteobacterial DNA in the blood of healthy volunteers, without positive bacterial culture, also supports a possible transient gut translocation through the intact gut barrier due to the invasiveness of pathogenic bacteria as normal physiology [80,81]. Here, COVID-19-induced gut dysbiosis was not prominent than the control group, as only *E. coli* DNA in severe cases was higher than the less severe infections, which was different from previous studies ([82,83], possibly because of the exclusion of the cases with prominent sepsis (hemodynamic alterations). On the other hand, increased BG in the serum of COVID-19 cases was similar to the findings of a previous publication [38], suggesting a possible influence of gut fungi. While serum BG levels were not well correlated with COVID-19 severity (BG levels were elevated independently of COVID-19 severity), endotoxin levels were a better determinant of disease severity. Both endotoxin and BG, as pathogen molecules, may, nevertheless, cause inflammation in a synergistic manner [9].

### 4.3. Synergy of Endotoxemia- and Glucanemia-Induced NETs in COVID-19, an Impact of Gut Barrier Defect

Serum of patients with COVID-19 (moderate and severe cases) activated NETs in isolated neutrophils from healthy volunteers, implying the impact of some blood components, such as pathogen molecules (LPS, BG, and bacteria-free DNA) and/or cytokines, in the samples [68,84,85]. After incubation by the serum of severe COVID-19 cases, NETs formation was even higher than the activation by PMA, a potent NETs inducer [9]. Due to the higher potency of LPS over bacterial DNA for inflammatory activation [42,86,87], only LPS was selected as a representative bacterial molecule for the in vitro experiments. As such, combined LPS with BG (LPS + BG) induced more prominent NETs than activation by each molecule alone, supporting the LPS-BG additive effect; however, the activity was lower than that of the PMA-positive control. The stronger NETs activation from the serum of severe COVID-19 cases in comparison with the activation by LPS + BG was possibly due to cytokines in the serum. Although all activators (PMA, LPS, BG, and LPS + BG) effectively induced NETs formation, only LPS and LPS + BG elevated cytokines. Meanwhile, PMA induced the highest level of ROS. These data implied that LPS and LPS + BG might induce NETs through cytokine mediation [43,44], while PMA facilitated NETs through ROS production [88]. Furthermore, BG slightly upregulated *NFκB*, but did not induce cytokines and ROS, supporting the idea that BG induces NETs through a Dectin-1-dependent mechanism, but not through cytokines or ROS [89]. Despite the presently limited exploration of the mechanisms of the LPS-BG synergistic NETs formation in COVID-19, common downstream signaling from TLR-4 and Dectin-1—such as spleen tyrosine kinase (SYK), extracellular signal-regulated kinases (ERK), protein kinase C (PKC), and nuclear factor kappa B (NFκB)—might be responsible for the synergy [90,91,92,93,94] (Figure 7). Moreover, nicotinamide adenine dinucleotide phosphate (NADH) activation by PKC also induces ROS and NETs-associated factors (MPO, NE, and *PAD-4*) [95] together with NFκB signaling [96] (Figure 7). For the working hypothesis, SARS-CoV-2 pneumonia, especially with hypoxia, induces inflammatory mediators, partly through pulmonary dysbiosis [97,98], together with the possible SARS-CoV-2 direct enterocyte infection [99]. Then, all these factors lead to gut barrier defects, allowing gut translocation of LPS and BG, which enhances a more severe systemic inflammation through the formation of NETs (Figure 7). Basic knowledge of the pulmonary–intestine axis, leaky gut, and NETs formation in COVID-19 might lead to a new treatment modality for sepsis.

### 4.4. Clinical Benefits and Study Limitations

For clinical benefit, our data shed light on the impacts of gut barrier defect, endotoxemia, glucanemia, and NETosis in severe COVID-19 infection. Hence, the detection of gut barrier defects using several indicators—including serum zonulin, the lactulose-to-mannitol excretion ratio, microbial molecules in serum without systemic infection (LPS and BG) [15,100], and/or NETosis determination—in patients with COVID-19 can indicate the disease severity. The laboratory procedures for NETs determination could be as simple as the neutrophil separation procedure, using density gradient centrifugation (the separation of sample components on the basis of the density difference of each component), with the identification of the DNA branching by DNA staining color (see method section), which would only need a centrifugation machine and fluorescent microscopy equipment. These machines are currently available as routine instruments in regular laboratory hospitals, used for anti-nuclear antibody detection. Otherwise, other parameters with more expensive costs can be also performed (such as determining histone–DNA and MPO–DNA) for the detection of NETs formation. Further study, presenting better cost-effective tests of NETosis, will be beneficial.

In addition, several novel biomarkers for differentiating the disease severity of COVID-19 were also studied, for example, serum-neuron-specific enolase (NSE—a pulmonary injury biomarker in lung cancer) [101], and presepsin (a soluble CD14; a proinflammatory activator of immune cells) [102]. These biomarkers might be indicators of the differences between some specific characteristics of the patients that might be required for determination. As such, while serum NSE indicates pulmonary involvement in COVID-19 [101], serum presepsin detects the early onset of sepsis [102]. Similarly, the detection of the gut permeability defect (or gut-pulmonary axis) might be used to alert physicians to developing severe systemic inflammation in COVID-19 cases, while NETosis biomarkers indicate an initial phase of the systemic responses. Moreover, the combination of these biomarkers might also be used to follow the progression of the disease severity from pulmonary involvement (serum NSE) to the enhanced systemic inflammation (gut permeability defect and NETosis) and/or the progression into sepsis (presepsin). Additionally, the neutralization of these parameters (alone or in combination)—including (i) the strengthening gut integrity using probiotics, short-chain fatty acid, and/or zonulin alteration [103,104]; (ii) LPS absorption [105]; and (iii) BG removal [106]—might be the new adjuvant treatment. Further studies on these topics are required.

Finally, several limitations in our study needed to be mentioned. First, the number of samples is limited, with only a cross-sectional evaluation at the time of hospital visitation. Additionally, the age and gender of the healthy volunteers are different from those of the patients. Timepoint evaluation of several parameters and improved recruitment of healthy controls will be informative. Secondly, samples from patients with apparent sepsis were excluded; therefore, the results might not apply for COVID-19 cases with sepsis, which is common. Although LPS and BG in serum of patients with severe sepsis from gut translocation have been demonstrated [30], neutropenia in severe sepsis is also common [107], which might decrease the importance of NETosis in COVID-19 cases with sepsis. Thirdly, SARS-CoV-2 strains are continuously mutating and the required responses to mutated viruses might be different [108]. More studies on this topic are warranted.

In conclusion, gut translocation of pathogen molecules (LPS and BG) and NETosis in severe COVID-19 cases are potential targets for biomarker development and new therapeutic modalities (leaky gut attenuation, PAMPs neutralization, and NETs inhibition) that will improve the clinical outcomes of the patients.

## Figures and Tables

**Figure 1 cells-11-01103-f001:**
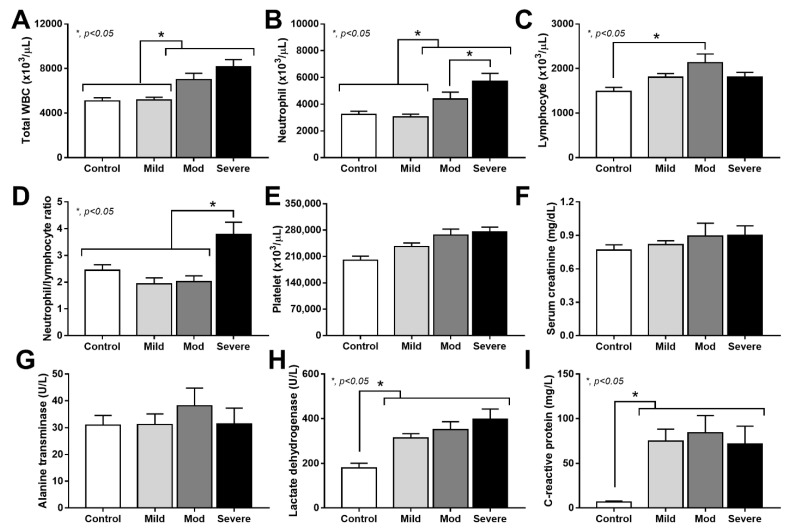
Characteristics of patients with COVID-19 in different severities—mild (*n* = 27), moderate (*n* = 28), and severe (*n* = 20), versus healthy control (*n* = 15), as indicated by total peripheral white blood cell count (WBC) (**A**), blood neutrophils and lymphocytes (**B**,**C**), neutrophil/lymphocyte ratio (**D**), platelet count (**E**), serum creatinine (**F**), alanine transaminase (ALT) (**G**), lactate dehydrogenase (LDH) (**H**), and C-reactive protein (CRP) (**I**)—are demonstrated. *—*p* < 0.05 between the indicated groups as determined by ANOVA with Tukey’s analysis.

**Figure 2 cells-11-01103-f002:**
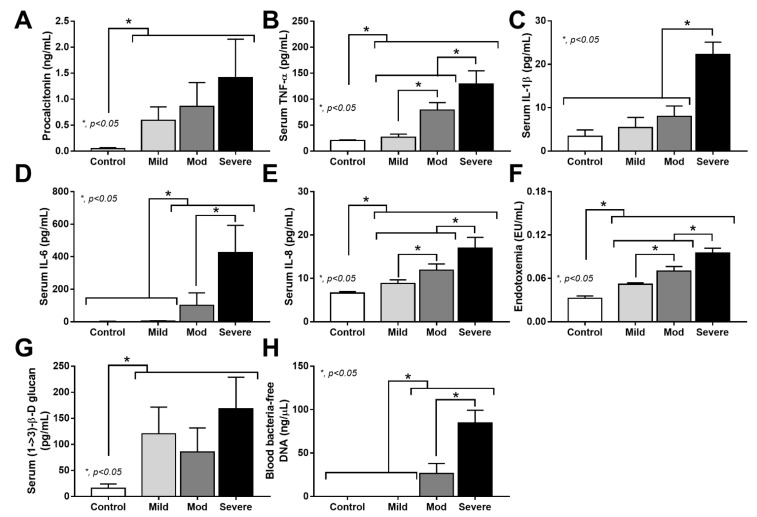
Characteristics of patients with COVID-19 in different severities—mild (*n* = 27), moderate (*n* = 28), and severe (*n* = 20), versus healthy control (*n* = 15), as indicated by procalcitonin (**A**), serum cytokines (TNF-α, IL-1β, IL-6, and IL-8) (**B**–**E**), endotoxemia (**F**), serum (1→3)-β-D-glucan (**G**), and blood-bacteria-free DNA (**H**)—are demonstrated. *—*p* < 0.05 between the indicated groups as determined by ANOVA with Tukey’s analysis.

**Figure 3 cells-11-01103-f003:**
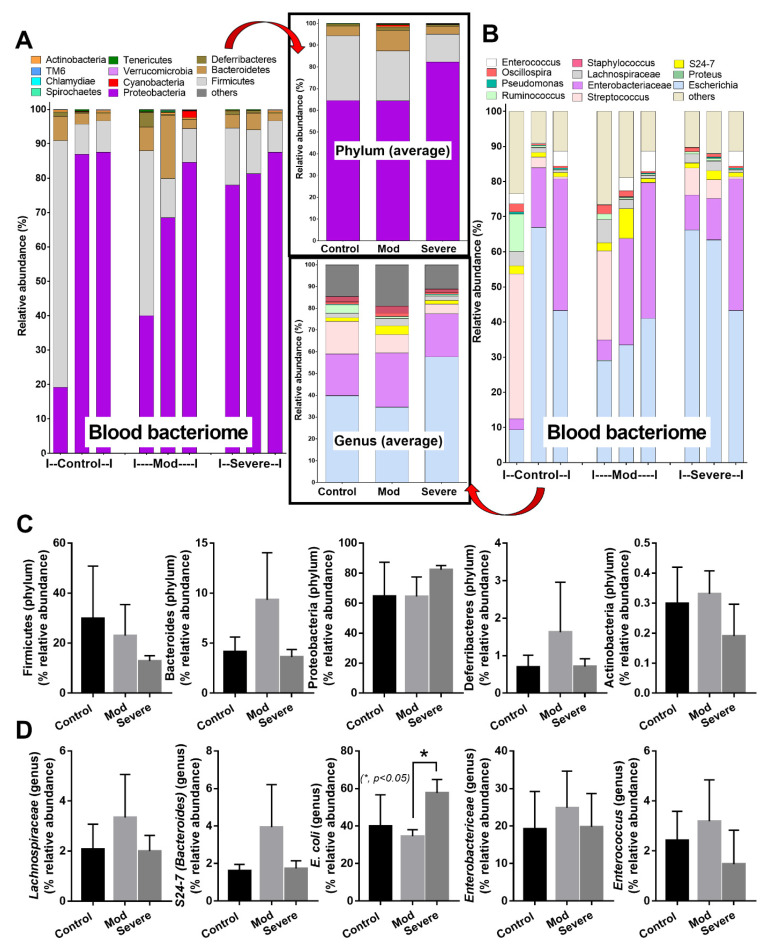
Blood microbiome analysis (blood bacteriome) from samples of healthy control (Control) and COVID-19 infection in moderate (Mod) and severe (Severe) conditions, as indicated by the phylum level analysis (with average value) (**A**), genus-level analysis, and average value (**B**). Graph presentation in phylum (**C**) and genus (**D**) analysis are demonstrated (*n* = 3/group). *—*p* < 0.05 between the indicated groups using one-way ANOVA with Tukey’s analysis.

**Figure 4 cells-11-01103-f004:**
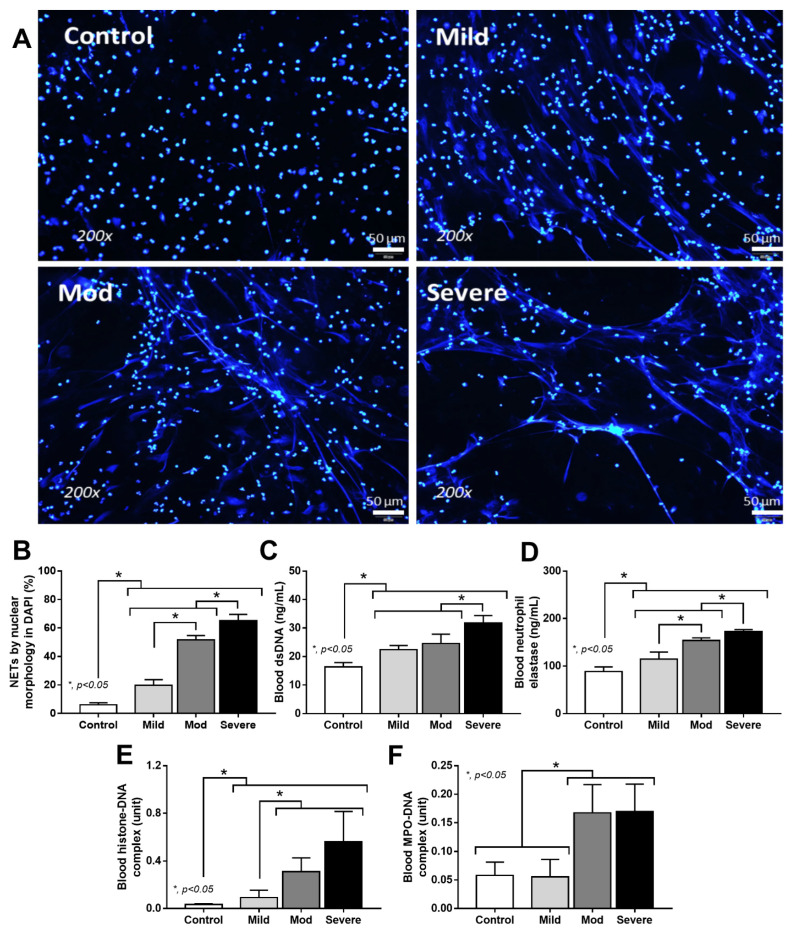
Neutrophil extracellular traps (NETs) of patients with COVID-19 in different severities—mild (*n* = 27), moderate (*n* = 28), and severe (*n* = 20), versus healthy control (*n* = 15), as indicated by morphology of DAPI nuclear staining with the representative pictures (**A**,**B**), blood dsDNA (**C**), blood neutrophil elastase (**D**), histone–DNA complex (**E**), and myeloperoxidase (MPO)–DNA complex (**F**)—are demonstrated. * *p* < 0.05 between the indicated groups as determined by ANOVA with Tukey’s analysis.

**Figure 5 cells-11-01103-f005:**
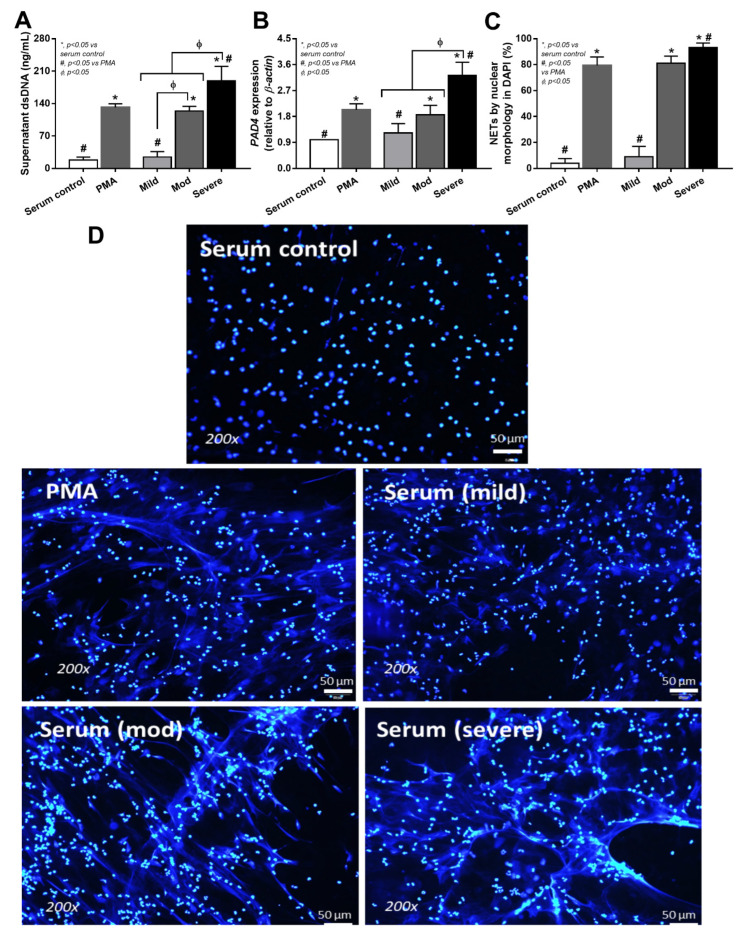
Neutrophil extracellular traps (NETs) of the isolated neutrophils from healthy volunteers after activation by control serum and serum from cases with different COVID-19 severities, using Phorbol 12-myristate 13-acetate (PMA) as a positive control of NETs inducers, as indicated by supernatant dsDNA (**A**), Peptidyl Arginine Deiminase 4 (*PAD-4*) expression (**B**), and morphology of DAPI nuclear staining with the representative pictures (**C**,**D**) (the cells from 7 volunteers were used in all activations). * *p* < 0.05 vs internal serum control groups, ^Φ^
*p* < 0.05 between the indicated groups and ^#^
*p* < 0.05 vs PMAas determined by ANOVA with Tukey’s analysis.

**Figure 6 cells-11-01103-f006:**
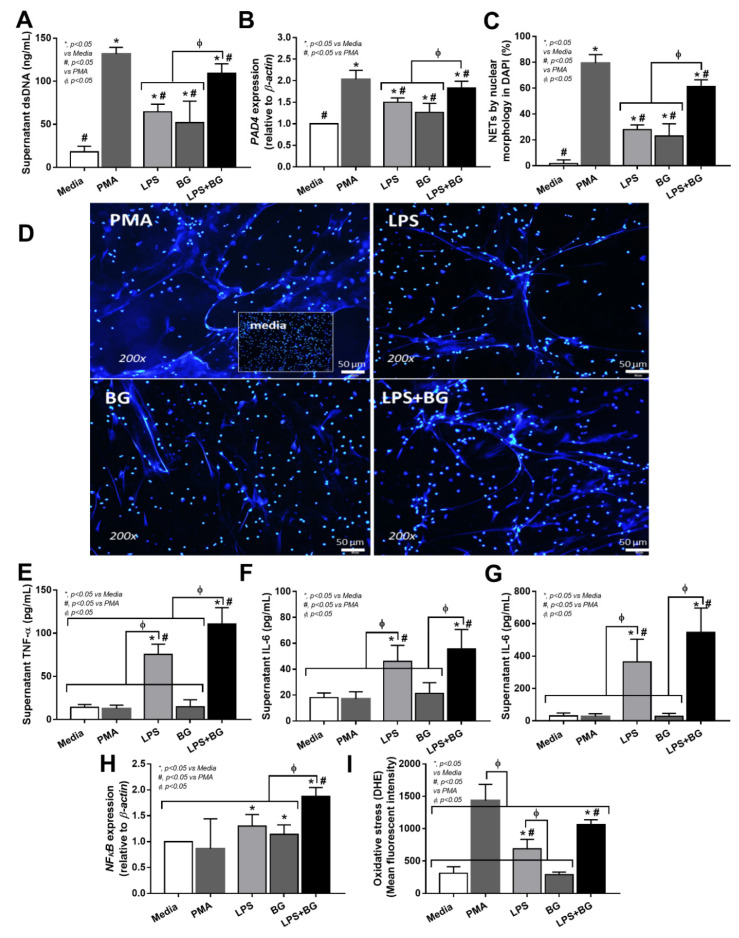
Neutrophil extracellular traps (NETs) of the isolated neutrophils from healthy volunteers after activation by control serum, lipopolysaccharide (LPS) with and without (1→3)-β-D-glucan (BG), using Phorbol 12-myristate 13-acetate (PMA) as a positive control of NETs inducers, as indicated by supernatant dsDNA (**A**), Peptidyl Arginine Deiminase 4 (*PAD-4*) expression (**B**), morphology of DAPI nuclear staining with the representative pictures (**C**,**D**), supernatant cytokines (**E**–**G**), nuclear factor kappa B (*NFκB*) expression (**H**), and reactive oxygen species (ROS) using dihydroethidium (DHE) (**I**) (neutrophils were isolated from 7 volunteers and each groups of neutrophils were used in all group of activations). * *p* < 0.05 vs internal serum control groups, ^Φ^
*p* < 0.05 between the indicated groups and ^#^
*p* < 0.05 vs PMAas determined by ANOVA with Tukey’s analysis.

**Figure 7 cells-11-01103-f007:**
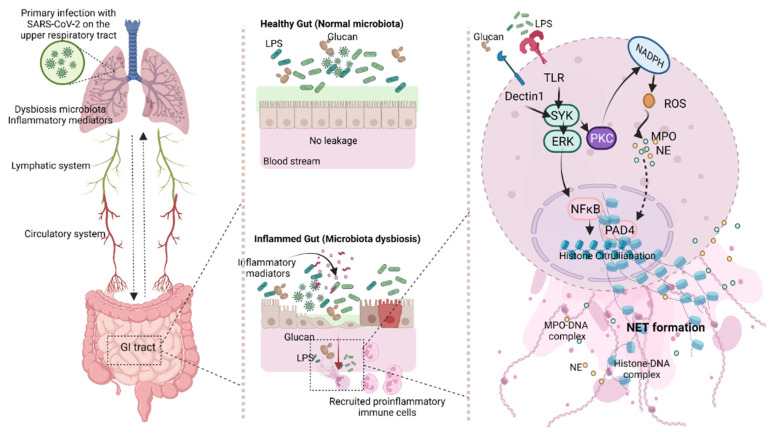
The working hypothesis proposes a mechanism of gut barrier defect and neutrophil extracellular traps (NETs) in SARS-CoV-2 infection that originates from the lung before infecting other ACE-2 positive cells, including enterocytes. Cytokine storm and enterocyte damage enable the gut translocation of pathogen molecules, lipopolysaccharide (LPS), and (1→3)-β-D-glucan (BG), that possibly enhance neutrophil responses through TLR-4 and Dectin-1, respectively, with downstream signaling, spleen tyrosine kinase (SYK), extracellular-signal-regulated kinases (ERK), protein kinase C (PKC), nuclear factor kappa B (NFκB), and nicotinamide adenine dinucleotide phosphate (NADH). These mediators finally activate NETs-associated molecules, peptidylarginine deiminase 4 (*PAD-4*), myeloperoxidase (MPO), and neutrophil elastase (NE). Gut barrier defect, cytokine storm, and neutrophilia in COVID-19 infection elevate systemic inflammatory responses and sepsis severity. ROS—reactive oxygen species.

**Table 1 cells-11-01103-t001:** The epidemiological data.

	SAR-CoV-2 Cases				Healthy Control (*n* = 15)
	Total	Mild	Moderate	Severe	
	(*n* = 75)	(*n* = 27)	(*n* = 28)	(*n* = 20)	
Age (year old)	38 ± 15	38 ± 11	36 ± 17	48 ± 19	31 ± 8
Gender—male, *n* (%)	48 (64)	19 (70)	12 (43) *	15 (75)	5 (33) *
Diabetes mellitus, *n* (%)	20 (27)	4 (15)	6 (21)	10 (50) *	0
Hypertension, *n* (%)	14 (19)	1 (4)	1 (4)	12 (60) *	0
Dyslipidemia, *n* (%)	12 (16)	0	0	12 (60) *	0
Lupus, *n* (%)	3 (4)	0	0	3 (15) *	0
CAD/CVA, *n* (%)	10 (13)	0	3 (11)	7 (35) *	0
COPD, *n* (%)	6 (8)	0	0	6 (30) *	0
Asthma, *n* (%)	2 (3)	0	0	2 (10) *	0
Malignancy, *n* (%)	1 (1)	0	0	1 (5) *	0
No known underlying disease, *n* (%)	47 (63)	12 (44)	4 (14)	0 *	15 (100)

CAD/CVA—coronary artery disease/cerebrovascular disease (conditions that increased atherosclerosis risks); COPD—chronic obstructive lung disease; *—*p* < 0.05 vs. other groups.

## Data Availability

The data are available from the corresponding author upon reasonable request.

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
