# Peer review of "Neutrophil Extracellular Traps in Severe SARS-CoV-2 Infection: A Possible Impact of LPS and (1→3)-β-D-glucan in Blood from Gut Translocation"

_cells, 2022, doi:10.3390/cells11071103_

Round 1

Reviewer 1 Report

The manuscript by Saithong S, et al aims to address the role of gut translocation, particularly from LPS and BG in NET induction in COVID19 pathogenesis. Authors found that LPS and BG act sinergistically to induce NETs, as well as the production of pro-inflammatory cytokines, and ROS. The methodologic approach is thorough regarding the bacterial proteam analyses, as well as the NET induction experiments, since all were confirmed by different techniques, including immunofluorescence.

There is a major issue to be addressed by authors since they used a COVID19 severity scale only based on local data. As previously described by other groups, most of the findings are dependent upon COVID19 severity. Therefore, authors should reclassify patients based on universally accepted definitions of COVID19 severity (ie Liu W, Tao ZW, Lei W, Ming-Li Y, Kui L, Ling Z, et al.. Chin Med J (Engl) (2020) 133(9):1032–38.; Living guidance for clinical management of COVID-19: Living guidance, 23 November 2021 – World Health Organization (WHO)) to analyze data and confirm their findings. 

Reviewer 2 Report

Saithong et al. submitted an interesting study providing evidence for the linkage of NET formation to gut barrier defects in patients with COVID-19. The strengths of this study are the analyses of the parameters and the differences found for patients with different severity of COVID-19. Also, the aim of the study, to link gut barrier defects with NET formation in these patients is highly interesting and relevant to the field to gain a better understanding and develop new treatment options. The following drawbacks should be attended:

  1. The materials and methods section should be a bit more precise to ensure reproducibility of these data by other groups, e.g. it should be mentioned which machines were used for the development of ELISAs etc. and how exactly the ROS measurement was performed (incubation time of the compound, detection of ROS production etc.).
  2. In the methods section it is also written that NETs were detected amongst others by MPO-DNA complex ELISA, however, in the manuscript and also in the figure legend for figure 4 and Figure 4F it is written histone-MPO complex. Which one is it? Histone-MPO or DNA-MPO?
  3.  Since neutrophils are highly variable between different individuals, there is concern about the healthy controls used since the mean age and the gender distribution do not match the patients’ data really well.
  4. The authors state in line 264-266 that the data implied that the gut translocation in healthy individuals only detectable by bacteriome analysis but not conventional method supported the possible physiologic gut translocation. The concern here would be that the conventional method is maybe not good enough to detect trace amounts of bacteria in the blood. How can the authors make sure that what they are detecting by bacteriome analysis is from the gut due to a gut barrier defect and not blood-borne but not detected by the conventional method?
  5. The authors themselves mention in the second last paragraph (lines 420-426) that they excluded patients with apparent sepsis, the question is, wouldn’t it be interesting to include these patients as a separate group to actually test the hypothesis as COVID-19 with sepsis is quite common.
  6. There is a mistake with the figure labelling since figure 7 is missing.
  7. Figure 8 is a nice summary of the work presented, however it should be mentioned and described in the text.
  8. In line 248 the authors reference Figure 2F for endotoxemia but Figure 2F shows blood bacteria-free DNA, I think it should be Figure 2D instead.
  9. Scale bars are missing in Figure 4A and Figure 5D. 100x is not enough.
  10. Serum control in Figure 5D should be as big as the other pictures, otherwise it is hard to see.
  11. The outlook should be a bit extended. What is the possible benefit of the results obtained from this study?
  12. It should be cleary stated that the patients also gave informed written consent. Why was the informed consent statement put as not applicable?
  13. The data availability statement is a bit short and unprecise, there is for sure related data especially concerning the patients that was not used for publication.

Reviewer 3 Report

dear Authors, I have read the manuscript and I send you my comments:

1) Methods and Results: "Confirmed cases of COVID-19 defined as positive for SARS-CoV-2 RNA using real time reverse transcription-polymerase chain reaction (RT-PCR) test" PLEASE ADD IN RESULTS THE CT VALUES in order to define the number of copy for this virus.

2) Methods and Results: please add data for other cytokines and chemokines and also for receptors involved in cells translocation.

3) Discussion: the authors suggest that NETs could be a biomarker but they must explain the method used for its isolation. A biomarkar must be easily isolated. Please explaine it.
4) A previous study documented that neuron specific enolase could be a biomarker of infection and it is related with severity of infection (PLoS One. 2021;16(5):e0251819. SCOPUS: 2-s2.0-85106356270) please discuss it. 

Round 2

Reviewer 1 Report

Authors have addressed all the concerns approppriately. 

Reviewer 2 Report

All my concerns were addressed in the revision and the language was significantly improved.

Reviewer 3 Report

Dear Auhtors,

thank you for your revisions, I have not further comments